# Retrospective Assessment of Thirty-Two Cases of Pelvic Fractures Stabilized by External Fixation in Dogs and Classification Proposal

**DOI:** 10.3390/vetsci10110656

**Published:** 2023-11-15

**Authors:** Jose Antonio Flores, Gian Luca Rovesti, Lucia Gimenez-Ortiz, Jesus Rodriguez-Quiros

**Affiliations:** 1Hospital Veterinario IVC Evidensia Prïvet, Calle Duero 37, Villaviciosa de Odón, 28670 Madrid, Spain; 2Clinica Veterinaria M. E. Miller, Via della Costituzione 10, 42025 Cavriago, Italy; gl.rovesti@clinicamiller.it; 3Clínico Veterinario Val de Iglesias, Calle de la Corredera Alta 30, San Martín de Valdeiglesias, 28680 Madrid, Spain; luciagimortiz@gmail.com; 4Departamento de Medicina y Cirugía Animal, Facultad de Veterinaria, Universidad Complutense de Madrid, Avda. Puerta de Hierro s/n, 28040 Madrid, Spain; jrquiros@ucm.es

**Keywords:** external fixation, external fixation classification, fractures, pelvis, dog

## Abstract

**Simple Summary:**

This study was aimed to evaluate the use of external fixation to stabilize pelvic fractures. The technique was performed by minimally invasive procedures, substantially reducing the impact of open approaches on tissues and facilitating healing of tissues. External fixation applied to pelvic fractures proved to be a valuable complimentary fixation method as well, providing further stability to the primary one. Therefore, though it is a technique not usually used in pelvic fractures, in our case cohort, it proved to be an effective alternative to the sole use of plates, either as an exclusive stabilization method or as an adjuvant one. During the evaluation of the cases, the authors were prompted to establish a classification proposal for the frame configuration used, in order to better understand the differences among them.

**Abstract:**

The goals of this study were to evaluate the outcomes of bone healing, patient comfort during the treatment, functional results, and complications in pelvic fractures treated with external fixation, as well as to propose a classification system for the applied external frames. A total of thirty-two canine patients with pelvic fractures of different origins were treated. To provide a better reference for the frames used, an alphanumeric classification system was developed, detailing the frame structure and the number and location of the pins used. In this study, eighty-six fractures were treated in the 32 patients of this work, with an average fixation time of 9.88 ± 4.15 weeks. No major complications were detected in this case cohort, and the outcomes were rated at 9.46 based on a visual assessment scale for the patient’s comfort during treatment. Outcomes graded as excellent and good were 96%. The use of external fixation for stabilization of pelvic fractures should be considered as a technical option, especially for minimally invasive stabilization of complex fractures, either as a primary or secondary stabilization.

## 1. Introduction

Pelvic fractures in small animals account for 25% of those diagnosed in daily clinical practice [1,2]. Most fractures in dogs are caused by traffic accidents, falls, or blows. Pathological fractures are less frequent [3]. Conservative treatment is an option that, although some studies have reported successful outcomes [4], often leads to inadequate reduction and needs a long recovery time [5]. Surgical treatment is generally considered the standard choice in most cases, providing a shorter recovery time, faster healing, a reduced incidence of diseases related to the stenosis of the pelvic canal, and a better functional outcome [6].

In human medicine, the acute management of pelvic fractures by external fixation (EF) is commonly performed to avoid displacement of the fracture site, bleeding, neurovascular damage and/or post-traumatic pain [7]. Additionally, EF is a valuable technique for definitive stabilization due to its low rate of complications and ease of application [7,8,9].

In the veterinary literature, some studies have been published in which EF was used for the treatment of pelvic fractures. The main characteristics of this technique are safety, ease of application, and patient compliance, generally providing a good outcome with a low rate of complications [4,10,11]. The most frequent complications encountered were loosening of the frame and local infection at the pin sites [4,12].

The present study describes the surgical technique and evaluates the outcome of the treatment of pelvic fractures using EF, either as the primary or as a complementary technique, in 32 canine patients.

The hypothesis underlying this study was that external fixation could offer a significantly stronger lever arm compared to what could be achieved with internal fixation alone, due to its extension, and with a much less invasive surgical approach. These characteristics were expected to ensure better stability of the treated fracture and better comfort for the patient during the postoperative (PO) period, whether external fixation was used as a secondary support technique to internal fixation or used as a primary stabilization technique. Additionally, this study introduces a classification system for the frame constructs used, due to the absence of nomenclature and classification for this type of EF in the existing literature.

## 2. Materials and Methods

In the present study, 32 dogs treated at Hospital Veterinario IVCE Evidensia (Villaviciosa de Odón, Madrid, Spain) and Clinica Veterinaria Miller (Cavriago, Italy) between 2006 and 2022 were reviewed. The inclusion criteria for this study were dogs with one or more pelvic fractures. In some patients, these fractures coexisted with extrapelvic injuries of various kinds. The critical condition of most dogs, mainly of a polytraumatic nature, was considered as an inclusion criterion, because of the minimal invasivity and shorter anesthetic time required for EF compared with plating. In other cases, EF was chosen as a complementary system to increase the stability of the fracture site treated primarily with internal osteosynthesis. For each case, the collected information were signalment, the nature of the injury, a description and latency of the surgical procedure, and fixation time. During the PO period, the surgeon in charge evaluated the patient’s comfort based on a visual assessment scale (VAS) ranging from 1 (indicating the worst comfort) to 10 (indicating the best comfort). Furthermore, the final functional outcome was evaluated based on follow-up (FU) evaluations.

Radiological studies were reviewed to confirm the quality of fracture reduction and the fixation systems used. Overall, data regarding age, level of activity, weight, breed or the grade of displacement in the fracture site were taken in account at the beginning of the surgical process. Additionally, it is worth mentioning that usually an open approach and a combined fixation was preferred in fractures highly displaced and heavier patients, while a closed approach and ESF as the unique method of stabilization was preferred for lighter animals and lightly displaced fractures. The external fixators were arranged in a linear configuration and used in different arrangements based on the evaluation of the surgeon in charge of the case. The pins were inserted at the iliac wing and the ischiatic tuberosity and, sometimes, the neck of the ilium, following the safe corridors described later. The pins used were threaded from 1.2 to 3.5 mm in diameter. Sometimes, K-wires were added to the frame construct for specific requirements. These pins were connected to a 3 mm stainless steel connecting bar using Meynard steel clamps (Insorvet, Barcelona, Spain) or to 5 mm carbon bars with plastic clamps (Polilock radiolucent EF system—Ad Maiora, Cavriago, Italy).

There was a considerable range of configurations employed, leading the authors to introduce a classification system for describing the frame designs used in this cohort. This classification utilizes an alphanumeric notation that conveys information about the number of pins placed in each of the recommended locations and the connection between the assemblies of both hemipelves. The information is listed from left to right, looking to the fixator frame from above. In the proposed classification, each hemipelvis is divided into three segments (Figure 1):-Segment I: Iliac wing.-Segment II: Body of the ilium and acetabulum.-Segment III: Ischium.

Each segment is followed by a number indicating the pins inserted at that location.

In the alphanumeric description of both the left and right hemipelves, there is a mention of how the pins are interconnected with the connecting bars within each hemipelvis. These interconnections were utilized in five distinct frame designs, each denoted by a capital letter (Figure 2):Type T: A single bar interconnects the pins located on both iliac wings.Type C: The bars connect the pins on the iliac wing and ischiatic tuberosity of one hemipelvis to each other, and with the pin located on the wing of the contralateral ilium.Type O: The bars interconnect the pins of each hemipelvis around the perimeter.Type L: The bars connect the pins of one hemipelvis to each other and with the pin located on the contralateral ischial tuberosity.Type X: The bars interconnect the pins around the perimeter and with a cross connection between the pins located at the vertices of the quadrilateral.

An illustrative example of the alphanumeric description utilized in this study is as follows. In the configuration:“I2.II1.III1.X.I1.II0.III1.”, the alphanumeric code indicates the following:

On the left side (hemipelvis):Two pins have been inserted in the iliac wing.One pin has been placed in the body of the ilium.One pin has been inserted into the ischium.

On the right side (hemipelvis):One pin has been inserted into the iliac wing.There are no pins in the body of the right ilium.One pin has been placed in the right ischium.

All of these pins are interconnected using an X configuration, which represents a quadrilateral design with crossed bars between the contralateral ilium and ischium, as shown in Figure 3.

### 2.1. Preoperative Management

Each dog underwent preoperative X-rays examination. In some cases, when the correct evaluation of the fractures was difficult just based on standard radiographic examination, a CT scan was performed to provide a more comprehensive 3D assessment. Anesthetic protocols were planned regarding each patient’s condition, using a multimodal approach based in the administration of medetomidine hydrochloride at a dose of 20 μg/kg (Sedín, Vetpharma Animal Health, Barcelona, Spain) and methadone at a dose of HCl 0.5 mg/kg (Insistor, Richter Pharma AG, Wels, Austria) by intramuscular route, followed by induction of propofol at a dose of 3 mg/kg (Propofol Lipuro 1%, BBraun Melsungen AG, Melsungen. Germany) by intravenous route. The anesthetic inhalational maintenance was carried out with isoflurane (IsoVet, BBraun Melsungen AG, Melsungen. Germany). As part of the preoperative protocol and during the anesthetic induction, cefazoline (at a dose of 22 mg/kg IV [Cefazolina Normon 1 g. Laboratorios Normon, Tres Cantos, Madrid. Spain]) was administered by intravenously.

### 2.2. Surgical Technique

The patients were shaved throughout their dorsal lumbosacral and gluteal region, extending below the stifle joints. They were positioned in sternal recumbency on the surgical table with the hind limbs abducted and with abdominal support to keep the pelvis as horizontal as possible on the surgical table. In the cases where internal osteosynthesis was performed, the external stabilization was performed after the internal procedure. For cases involving only external stabilization was used, a fluoroscopy-guided reduction in the fracture was attempted by manipulating the pelvis and the ipsilateral limb if it was not luxated. In some cases, especially in heavier patients or older fractures, this procedure was challenging, and the reduction achieved was not accurate, which represents a limit inherent to closed reduction procedures.

Safe corridors for pin placement were identified for the placement of the pins. Threaded self-tapping pins (Screwpin—Ad Maiora—Cavriago, Italy) were inserted using a pre-drilling technique at 300–500 rpm. The technique for pin insertion in the ilial wing was as follows. The ilial wing is easily palpated on the lateral aspect of the lumbo-sacral area. A stab wound was made at performed on the cranial part of its craniocaudal length, and the bone is contacted with a sleeve inserted through it to avoid wrapping soft tissues around the drill bit and the pin’s threads. Depending on the patient’s size and weight, a pin of an appropriate diameter was inserted at an angle of 10–15° from proximo-lateral to disto-medial with respect to the vertical plane. The corridor for pin insertion in the body of the ilium was located in its caudal aspect and along the origin of the deep gluteus muscle. The pin was inserted at an angle of 45–90° with respect to the horizontal plane. The third corridor was located on the ischiatic tuberosity. A stab incision is made over the palpable area of the ischiatic tuberosity, and the pin was inserted at an angle of 10–15° from the vertical plane in a proximo-lateral to medio-distal direction. After the first two pins were inserted, fragment distraction was performed whenever possible, achieving the best possible reduction in the fragments through a closed approach by manipulating the pins in a joystick way with fluoroscopic assistance. In other cases, the fracture reduction was achieved through minimally invasive approaches. Sometimes, a smooth small-diameter K-wire was used for temporary stabilization of the fracture fragments to facilitate the insertion of threaded pins, which could either be removed or left for ancillary stabilization. When internal fixation was performed, the external fixator is always implanted after the internal osteosynthesis. Finally, the pins were connected using the clamps and connecting bars selected for each patient. Cefazoline (at a dose of 22 mg/kg) was administered by intravenously and repeated every 2 h during the entire procedure.

### 2.3. Postoperative Care

The duration of hospitalization time varied depending on the specific needs of each patient, which included the administration of fluids, antibiotics, anti-inflammatories, and analgesics. All owners were informed of the rationale of the treatment performed, explaining pros, cons, and information on PO care instructions.

Frequent dressings of the pin entry points and strict rest and confinement were recommended for the first few weeks of the postoperative period. Most owners were able to perform daily fixator care correctly at home. Subsequently, and until the removal of the fixator, owners were advised to engage in leash walks and controlled activity. The patient’s comfort during the time the fixator was in place was evaluated by the surgeon in charge weekly using a visual assessment scale (VAS). This scale ranged from 1 to 10, where a score of 1 indicated very poor comfort and 10 indicated excellent comfort for the patient.

### 2.4. Follow-Up

During the PO period, clinical and radiological examinations were performed at the attending surgeon’s judgment and based on the clinical progression of each case. Usually, the X-ray rechecks were scheduled at 3, 6, and 12 weeks after surgery, although this protocol was adjusted as needed based on individual patients. Differently from the usual radiographic standard, the sagittal projection was dorso-ventral because the patients cannot be placed in dorsal recumbency due to the presence of the fixator. In some cases, lateral oblique projections were used when overlapping existed. Concurrently, an overall evaluation of the fixator’s condition and PO care were carried out. Once bone healing had progressed sufficiently, the external fixator (EF) was removed under sedation.

The outcome of the treatment was assessed based on the fracture healing, with an outcome scale that considered the final functional result and residual pain. The grading scale used was as follows.

Excellent: No walking difficulties nor apparent pain.Good: Good functional result with residual lameness or mild signs of pain.Fair: Obvious and constant but not disabling lameness or signs of mild to moderate pain.Poor: Severe lameness or constant pain.

Complications related to the original trauma, for example of neurological origin, were included in this assessment, but not considered as complications related to the treatment.

## 3. Results

The stabilization of pelvic fractures was performed in thirty-two dogs. The mean age was 5.13 ± 4.27 years (median = 8.5 years; range = 0.15–14 years) and the mean weight was 14 ± 10.96 kg (median = 8.65 kg; range = 3–45 kg).

A total of 86 fractures were recorded in the 32 dogs of this study, and they were located as follows.

Ilium (*n* = 12; 14% [bilateral ilium: *n* = 4; 4%]); acetabulum (*n* = 10; 11% [bilateral acetabulum: *n* = 4; 4%]); pubis (*n* = 18; 21%); ischium (*n* = 20; 23%); sacroiliac dislocation (*n* = 26; 30% [bilateral sacroiliac dislocation: *n* = 16; 18%]). Car accidents (*n* = 25; 78%) were the most common cause, followed by falls or trauma of various origins (*n* = 6; 18%). In one case (3%), the cause was not determined. In this review, isolated ischial and pubic fractures were treated conservatively. Table 1 summarizes, in absolute numbers and percentages, the types of pelvic fractures treated and the EF configurations used in each case.

In 21 (65%) cases, a total of 27 lesions located in other sites of their skeletal system were found. The injuries associated with pelvic fractures were sacral fractures (*n* = 14; 51%); coxofemoral dislocation (*n* = 4; 14%), femur fractures (*n* = 3; 11%), tibia fractures (*n* = 2; 7%), fracture of the proximal femoral physis (*n* = 3; 11% [bilateral: *n* = 1–3%]); and fracture of the vertebral body of the 5th lumbar vertebra (*n* = 1; 3%).

In the stabilization of pelvic fractures, an EF system was used as the sole stabilization technique in 23 patients (71%). In 20 (87%) out of the 23, the mean fixator time was 9.88 ± 4.15 weeks (median = 8 weeks; range = 1.3–20 weeks). In the remaining three cases (13%), the fixator removal was performed by the referring veterinarian, and the time could not be determined exactly. A combination of external and internal osteosynthesis was used in nine cases (28%). The external fixator was used as a secondary stabilization together with plating in five (55%) fractures of the ilium and four (44%) of the acetabulum. In these patients, the mean fixator time was 8.28 ± 5.82 weeks (median = 8 weeks; range = 2–20 weeks). In one case, the time could not be determined exactly because the procedure was performed by the referring veterinarian.

In most cases, the reduction with closed pinning was performed with a fluoroscopy-assisted technique, while internal fixation required open access.

The total number of each type of EF used in this work is as follows: 13 type O (Figure 4, Figure 5, Figure 6 and Figure 7), 7 type C (Figure 8), 6 type T, 4 type X (Figure 9) and 2 type L.

Twelve fractures of the ilium were registered in 10 patients (31%). Four (40%) of these patients were treated exclusively with EF, and the healing period was 7.66 ± 0.57 weeks (median = 8 weeks; range = 1.3–8 weeks). In six (60%) patients, EF and plate were combined. In two of them, the time of bone consolidation could not be determined, because the removal of the implants was performed by the referring veterinarian. In the remaining four cases, the mean fixator time was 8 ± 3.65 weeks (median = 11.65 weeks; range = 4–12 weeks). The most common configuration in the 10 patients with iliac fractures was type O (*n* = 4), followed by type C and T (*n* = 2 each), and type X and L (one each).

The patient comfort graded by VAS was 8.75 ± 1.28 (median = 9; range = 7–10) in the 8 patients with unilateral fractures. In the bilateral cases, the comfort VAS was 10 in both with a type O configuration.

The outcome graded by VAS in the patients with unilateral fractures was excellent in five cases (62%), good in two (25%), and fair in one. Only mild neuropraxia of the pudendal nerve was detected in one case, and it resolved after the removal of the implant. Infection of the pin tracts was present in one case, and in another case the EF frame was too heavy and required changes to debulk it. The rest of the patients did not experience major complications due to the fixator. The outcome of the two dogs with bilateral iliac fracture was graded as excellent for both.

Eight patients (25%) with acetabular fractures were treated, and two of them were bilateral. Of the 10 acetabular fractures, 7 were treated exclusively with EF, and the healing period was 9.6 ± 5.44 weeks (median = 8 weeks; range = 5–20 weeks). Non-reducible acetabular fractures were treated with EF alone. The rationale of this choice was that if the procedure carried a too high risk of being unsuccessful, leaving implants that were not effective enough due to the fragmentation, it would not be helpful for the patient and could have prevented a hip prosthesis in a later phase. The types of configurations used in the resolution of unilateral acetabular fractures were O and X (*n* = 2 each). The other two patients (*n* = 2) were treated by combining the plate with EF, with a mean fixator time of 6.66 ± 2.30 weeks (median = 8 weeks; range = 4–8 weeks). In these cases, the types C and X (*n* = 1 each) were used. In the two cases with bilateral fractures, the O type was used, combining EF and plate in one of them. In our comfort VAS assessment, unilateral fracture patients were graded 9.33 ± 1.21 (median = 10; range = 7–10). In the two cases with bilateral fracture, the comfort VAS was graded 8 and 10. In one of these cases, femoral head and neck ostectomy had to be performed four months after fracture osteosynthesis due to femoral head necrosis. Histopathological analysis was not performed but a vascular damage due to the contusion was considered to be the most likely cause. The outcome was excellent for five (83%) patients with unilateral fracture. One of the six cases was valued as good because developed an L7-S1 discospondylitis. Bilateral fracture cases were graded as excellent and good. The latter one suffered a femoral and tibial fracture that required a long recovery time.

Ten cases (31%) of unilateral and eight (25%) of bilateral sacroiliac luxation were operated on. All bilateral injuries were stabilized with a sacroiliac screw and anti-rotational pin in addition to external fixation. The insertion of the anti-rotational pin was achieved once the dislocation was previously reduced and stabilized by the lag screw. The frame configurations were type O in five (62%), type T in two (25%), and type X in one. The mean fixator time was 8.5 ± 2.66 weeks (median = 8.5 weeks; range = 4–12 weeks). The stabilization for the 10 unilateral luxation was performed with EF in three cases with types C, L and T. Of the remaining cases, two were stabilized with crossed wires inserted between the iliac wings with type C external fixation. In the remaining five cases, the luxation was fixed using a screw and an anti-rotational K wire, with type O EF in three, type L and type C in one each. In patients with sacroiliac dislocation, type O was used in eight cases (44%), type C in four (22%), type T in three (16%), type L in two (11%), and type X in one. The overall mean fixator time was 8.37 ± 4.95 weeks (median = 8 weeks; range = 4–20 weeks). VAS for unilateral luxation was graded 9.6 ± 0.96 (median = 10; range = 7–10). In these cases, the outcome was excellent for eight patients (80%), good for one and fair for one. VAS for the eight patients with bilateral sacroiliac luxation was graded 9.6 ± 0.74 (median = 10; range = 8–10). The outcome was graded as excellent for six (75%), good and fair one case each. The overall result of comfort VAS for the 32 patients of this study was 9.46. Twenty-six (81%) patients had an excellent outcome, six (15%) were graded good and one fair.

The complications related to EF exclusively were as follows: loosening of the pins (*n* = 3–8%), occasional bleeding (*n* = 2–5%), and mild local infection processes at the pin insertion points (*n* = 15–39%). One patient suffered neuropraxia of the pudendal nerves due to the position of the fixator’s pins. The rest of the neurological damages were pre-existing to the treatment and caused by the original trauma. Isolated ischiatic and pubic fractures were treated conservatively.

## 4. Discussion

There are few studies in the literature addressing EF applied to pelvic fractures in small animals [5], in opposition to human medicine, where its use is common in unstable pelvic fractures, especially in the early stages of treatment [13,14], due to its known biomechanical properties and its peculiar characteristics [15,16,17,18,19]. The modularity of the system and its simple application, either as a temporary or definitive method for fracture resolution, provides speed as a surgical technique, i.e., offering notable advantages in critical patients [20]. These fixators can be radiolucent, thus greatly facilitating the evaluation of healing [21]. Radiographic rechecks are easier, avoiding the need for oblique projections as usual with standard metallic EF frames (Figure 4 and Figure 5). In addition, in patients treated with fluoroscopic-assisted techniques, the interference during the procedure is considerably reduced, allowing reduction maneuvers to be performed much more easily. However, a clear downside of the use of fluoroscopy is the radiation exposure, which may be considerable in long surgical procedures. Nevertheless, in our opinion, a highly experienced surgeon can reduce considerably the amount of radiation needed.

Another advantage for their use as a secondary technique, together with internal osteosynthesis as the primary technique [22,23,24,25], is represented by the added rigidity for the stabilization of unstable fractures [1,2,3,4,5,6,25,26,27]. This is mainly due to their lever arm, which can be comparable to the entire length of the pelvis, while the plate has usually a much shorter lever arm than the pelvis.

There is limited information available on the stiffness of the external fixation in comparison to internal in iliac fractures. Only one study concluded that the stiffness of both systems was similar, attributing notable biomechanical advantages to the external system due to its greater lever arm [5].

The outcome of unilateral ilium fractures was graded as excellent and good in 87% of the patients according to our assessment scale. The comfort VAS was 8.75 out of 10 for these eight cases, suggesting that EF was well tolerated by the patients. In most cases, the fixator time was almost the same of bone healing time. EF was used as the only stabilization technique in one of the two patients with bilateral fractures. In these two patients, the comfort VAS was graded 10 and the outcome was excellent for both. However, the progression of osteoarthritis is to be taken into account in the mid-long term when the fracture reduction is not anatomical, as is the case for most cases of EF. This can happen, though, also when anatomical reduction is not achieved by plating, and should be considered when planning an aggressive approach.

Most veterinary literature for the treatment of acetabular fractures in dogs reports the use of plates for stabilization [28,29]. Only Graville et al. (2018) described the use of EF for the stabilization of this type of fracture [11], suggesting that it was a viable option for the treatment of acetabulum fractures despite some limitations, such as challenging the reduction in the fracture site or pin loosening in some cases [5]. In our experience, the outcomes were good, with 67% of acetabular fractures having been treated exclusively with EF, achieving an acceptable time for bone consolidation. Seventy-five percent of the patients have graded a mean of 9.25 with comfort VAS. These results were considered satisfactory due to the specific features of the treated fractures and the reported critical management of acetabular fractures and their complications [28,30].

Among the registered complications, one dog underwent ostectomy of the femoral head and neck after undergoing fracture treatment with a plate and EF due to necrosis. Another patient developed an L7-S1 discospondylitis of unknown origin. The other complications registered were minor and had no significant impact on the expected outcome. The patients recovered a full clinical function of their affected pelvic limbs. Since it is not always possible to achieve an anatomical reduction with EF, and even though short-term functional recovery was excellent in most patients, osteoarthritis might develop in cases of non-anatomical reduction, particularly in patients with acetabular fragmentation. In those cases, though, the success of internal osteosynthesis was very debatable. The most used type of EF was O (62%), with pins inserted in both iliac wings and ischia.

In the present study, 26 sacroiliac luxations in 18 dogs were treated. The standard technique with screw and anti-rotational K wire to stabilize the sacroiliac joint [31,32,33,34] was applied in most cases. Only three fractures were exclusively treated with EF (11%), and they were unilateral. In the remaining 23 cases (88%), EF was employed as an additional technique to protect the internal osteosynthesis. In unilateral fractures, the support of the dislocated hemipelvis was provided by the external fixator anchored to the healthy hemipelvis. Type O was used in nine (50%) of the 18 patients, providing the necessary stability for an unstable type of injury, as reported in several studies [1,2,3]. The comfort VAS for all patients with sacroiliac luxation was graded 9.6, confirming the good acceptance of EF. Outcome was excellent for 14 (77%) out of 18 patients, which was considered satisfactory with a low rate of complications. Three patients with good outcome suffered mild miction and defecations disfunctions, which resolved after EF removal. Only in one patient was the outcome considered fair, primarily due to a multitude of complications related to other traumatic lesions. This large breed puppy suffered a vertebral L5 fracture with severe neurological deficit, in addition to a bilateral femoral capital physeal fracture, and for this reason it was not considered a primary complication of the EF application.

As previously mentioned, the fractures of the ischium and pubis were treated conservatively, avoiding the insertion of pins in the ischial tuberosities that were fractured.

One of the limits of the present study is its retrospective nature, because the philosophy and technical aspects of treatment changed over the course of this study, resulting in inconsistent treatment for all cases.

Another limitation is the difficult clinical assessment due to the pain and residual lameness encountered as part of the evaluation. We consider unavoidable to evaluate results under these conditions although a critical and objective analysis was performed at every moment. Additionally, making direct comparisons due to the highly varied conditions of patients at presentation may be challenging. This aspect, of course, was determined by the complexity of the fractures, and it is inherent to pelvic fractures, but also related to the concomitant injuries that are very common in dogs with severe trauma. For this reason, it is difficult to correctly evaluate the biomechanical behavior of coupling internal and external osteosynthesis techniques. A long-term analysis and biomechanical testing would be advisable for such evaluations.

## 5. Conclusions

In the study of 32 dogs presented here, EF has proven to be a viable option for the stabilization of pelvic fractures. The technique is safe provided that safe pin insertion corridors are followed, allowing fracture stabilization with minimal disturbance to the biological response. This characteristic minimizes the risks of iatrogenic damage [35,36] and infection while accelerating healing times [4,10,11]. In our experience, the outcome has been favorable when EF is used as a complementary fixation method, contributing to the holding power of internal osteosynthesis. When applied as the sole stabilization technique, EF has shown great efficacy in treating fractures of the ilium. EF alone was considered insufficient for the treatment of sacroiliac dislocations, due to the inherent instability of this type of injury [1,2,3], and internal fixation was considered necessary in most cases. Nonetheless, further biomechanical studies are required to provide a more comprehensive understanding of the stiffness and stability of various EF configurations in response to loads or specific applications. The evaluation of patient comfort with EF systems has shown a high level of acceptance among the dogs included in this study.

In conclusion, the proposed classification provides a useful method for better understanding the structure of the frames applied to pelvic fractures and facilitates the sharing information. However, and in the absence of a similar classification published in the bibliography, modifications or adjustments may be necessary in the future. So far, this classification has proven to be able to clearly describe the pelvic EF frames used in this study.

Overall, due to the ease of application of EF as a stabilization technique, the limited complications encountered, and the good tolerance of the system by the patients, EF should be considered among the techniques that can be used, whether as a primary or secondary approach, for stabilization of pelvic fractures in the dog.

## Figures and Tables

**Figure 1 vetsci-10-00656-f001:**
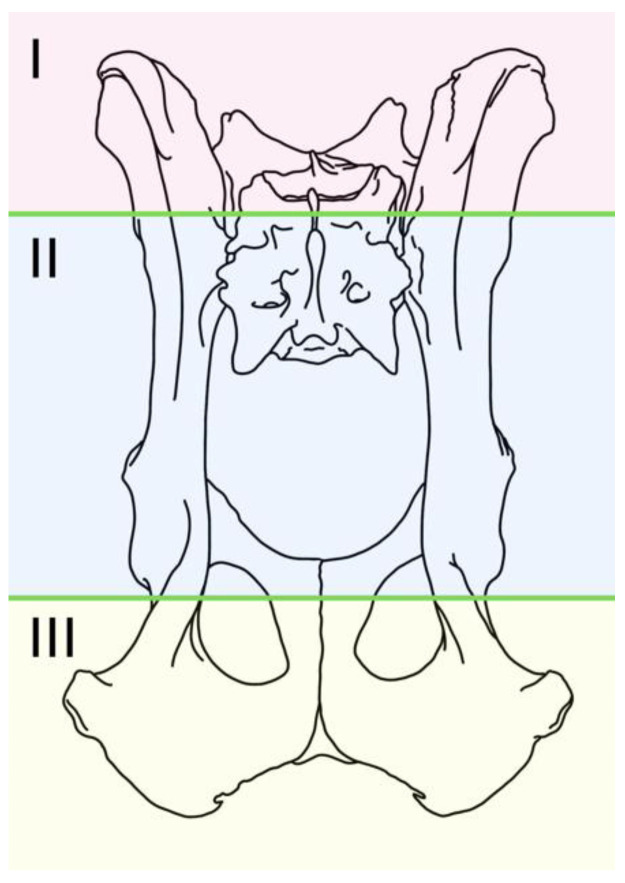
Segments into which the pelvis can be divided according to the proposed nomenclature for pelvic external fixation systems.

**Figure 2 vetsci-10-00656-f002:**
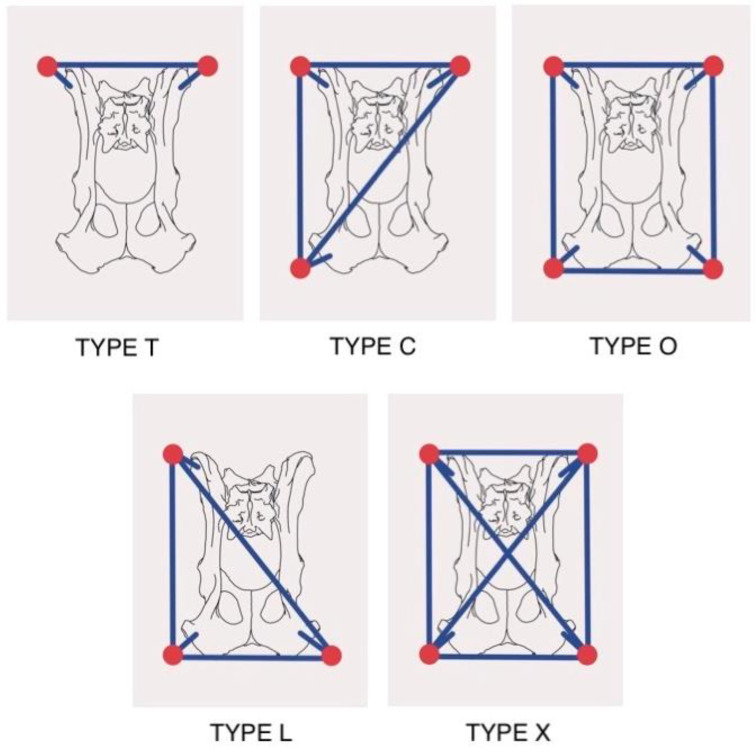
Different types of interconnections between hemipelves used as external fixation in the present study, as well as a classification proposal. Type T: a single bar interconnects the pins located on both iliac wings. Type C: the bars connect the pins on the iliac wing and ischiatic tuberosity of one hemipelvis to each other, and with the pin located on the wing of the contralateral ilium. Type O: the bars interconnect the pins of each hemipelvis around the perimeter. Type L: the bars connect the pins of one hemipelvis to each other and with the pin located on the contralateral ischial tuberosity. Type X: the bars interconnect the pins around the perimeter and with a cross-connection between the pins located at the vertices of the quadrilateral.

**Figure 3 vetsci-10-00656-f003:**
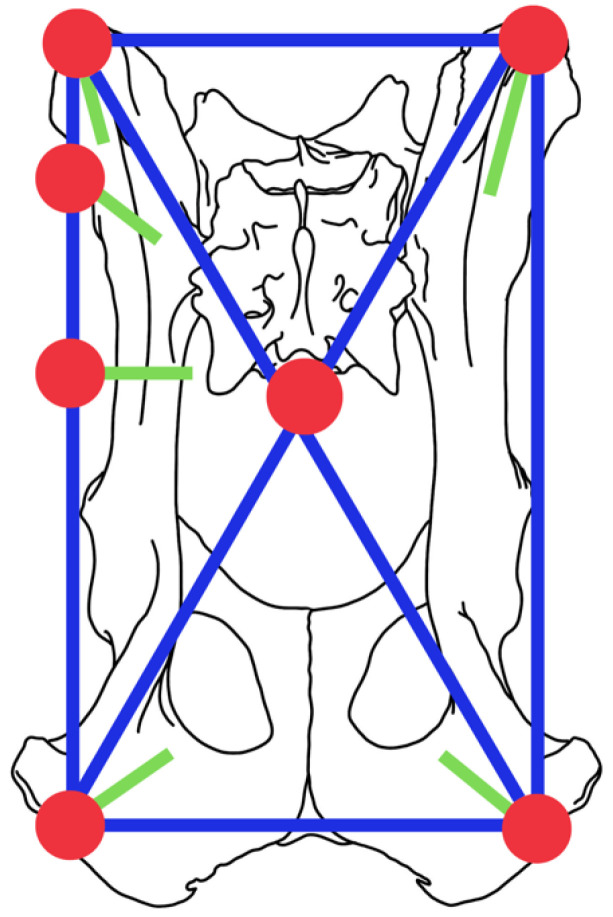
Descriptive illustration of the example for the proposed classification for pelvic external fixators with a configuration “I2.II1.III1.X.I1.II0.III1”.

**Figure 4 vetsci-10-00656-f004:**
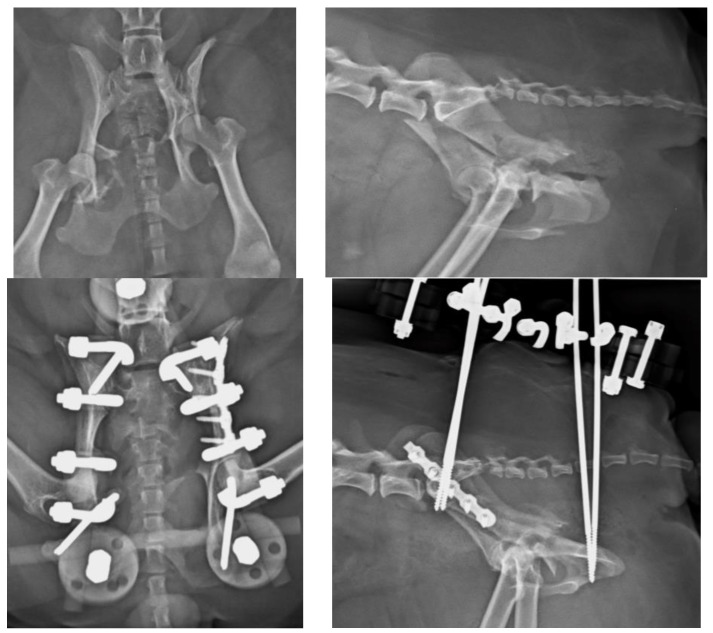
(**Top left**): Ventrodorsal (VD) projection of a patient with a fracture of the left ilium and a conminuted fracture of the contralateral acetabulum. (**Top right**): lateral projection of the same patient. (**Bottom left**): The fractures were reduced and stabilized by a type O radiolucent EF system. Note the plate used for the fracture of the ilium. (**Bottom right**): lateral projection of the same patient.

**Figure 5 vetsci-10-00656-f005:**
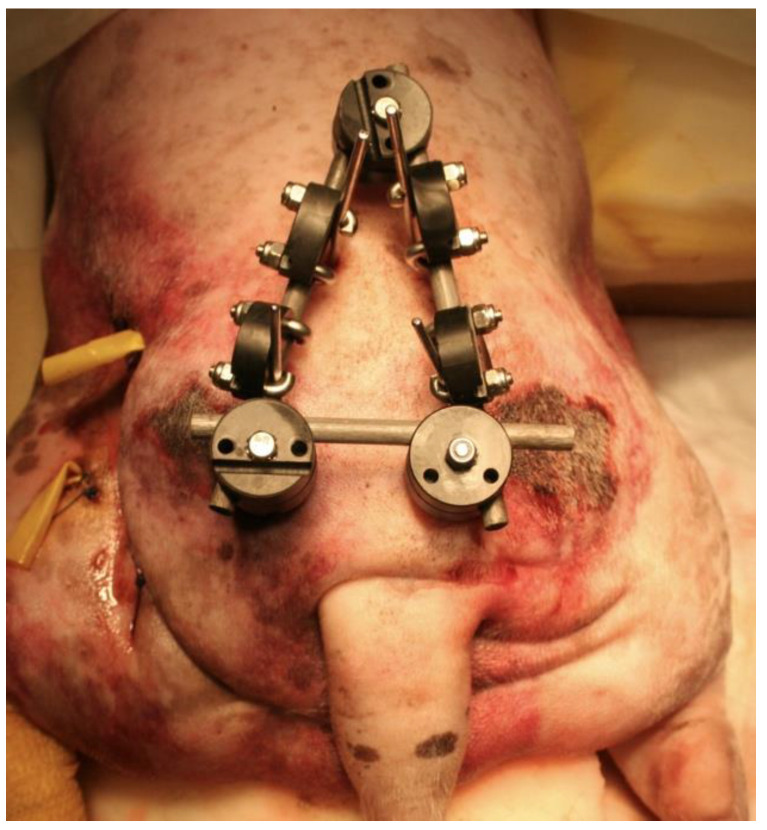
External appearance of the EF system used in the patient of Figure 4. A Polilock system composed of plastic and carbon fiber material was used.

**Figure 6 vetsci-10-00656-f006:**
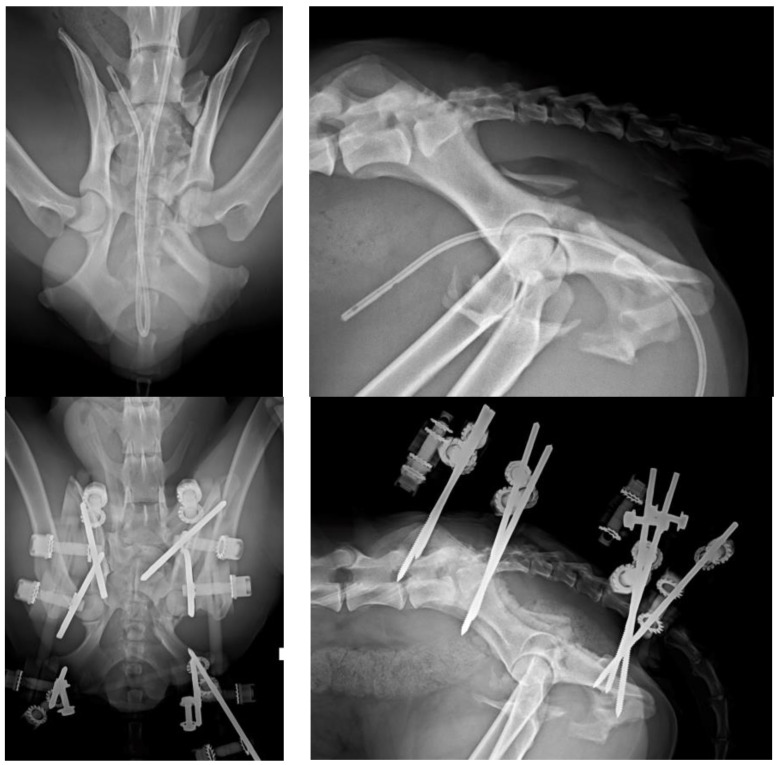
(**Top left**): VD projection of a patient with left acetabulum, ischial and pubic fractures. Note the fracture of the left articular process of the sacrum. (**Top right**): Lateral projection of the same patient. (**Bottom left**): The fractures were stabilized by a type O radiolucent EF. The radiolucent components of the EF provide a better visualization of the bone. (**Bottom right**): PO lateral projection of the patient.

**Figure 7 vetsci-10-00656-f007:**
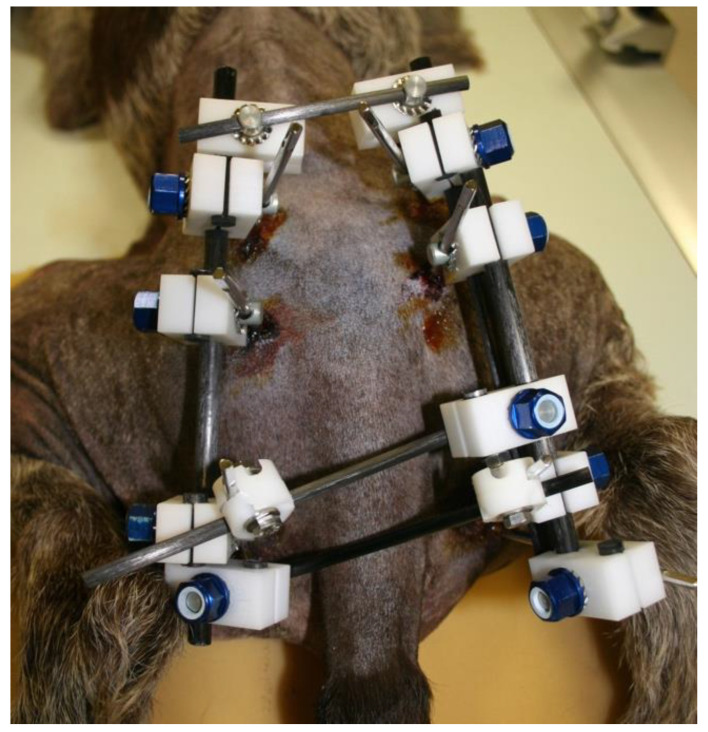
Picture of the type O EF used in the patient of Figure 6. Note the double connection bar between pins located in both ilial wings and ischia.

**Figure 8 vetsci-10-00656-f008:**
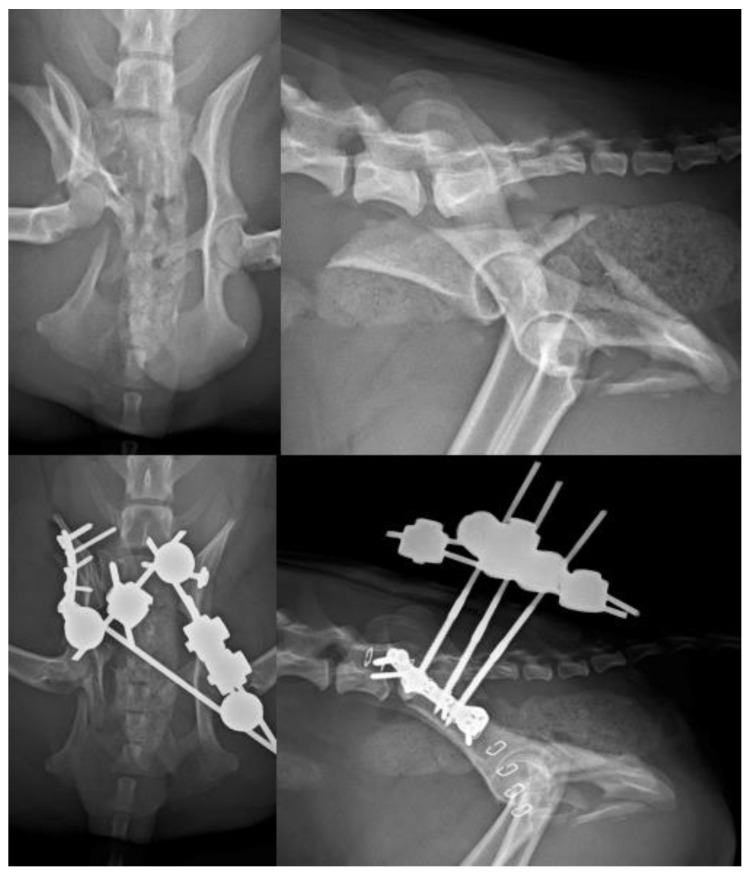
(**Top left**): VD view of a patient with fracture of the right ilium and of the right sacral articular process, and a left sacroiliac luxation. (**Top right**): Lateral projection of the same patient. (**Bottom left**): A combination of a type C EF and a plate have been used to stabilize the fractures. Note the lag screw for the left sacroiliac luxation. (**Bottom right**): Lateral projection of the same patient.

**Figure 9 vetsci-10-00656-f009:**
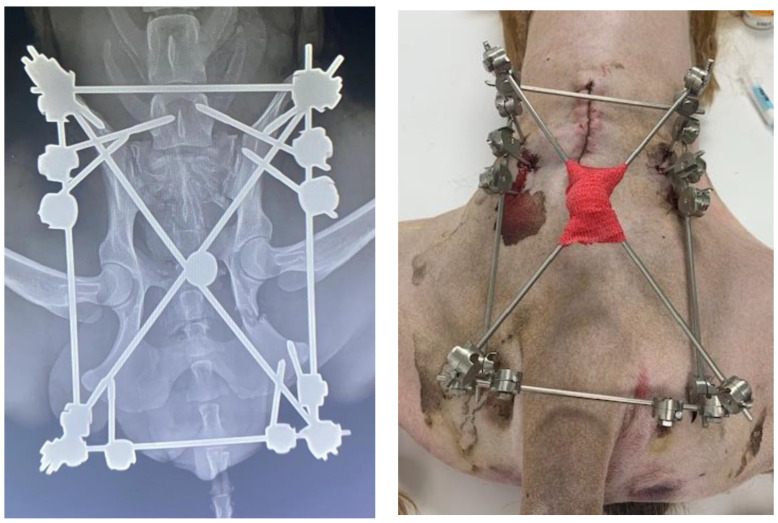
(**Left**): Radiograph of a patient treated with a type X EF. Note the fracture located on the ischial branch of both hemipelves and the pins inserted into the 7th lumbar vertebra to stabilize an additional fracture of the right sacral articular process. (**Right**): Picture of the external appearance of the EF.

**Table 1 vetsci-10-00656-t001:** Table showing, in absolute numbers and percentages, the type of pelvic fractures treated, and the external fixation configuration used. For ischiatic and pubic fractures the configuration used depended on another coexistent pelvic fracture. Isolated ischiatic and pubic fractures were treated conservatively.

	Ilium	Bilateral Ilium	Acetabulum	Bilateral Acetabulum	Sacroiliac Dislocation	Bilateral Sacroiliac Dislocation
Type O (*n* = 13)	2 (25%)	4 (100%)	2 (33%)	4 (100%)	3 (30%)	10 (62%)
Type C (*n* = 7)	2 (25%)	0 (0%)	1 (16%)	0 (0%)	4 (40%)	0 (0%)
Type T (*n* = 6)	2 (25%)	0 (0%)	0 (0%)	0 (0%)	1 (10%)	4 (25%)
Type X (*n* = 4)	1 (12%)	0 (0%)	3 (50%)	0 (0%)	0 (0%)	2 (12%)
Type L (*n* = 2)	1 (12%)	0 (0%)	0 (0%)	0 (0%)	2 (20%)	0 (0%)
TOTAL FRACTURES	8	4	6	4	10	16

## Data Availability

All data generated and analyzed during this study are included in this published article.

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
