# Peer review of "Retrospective Assessment of Thirty-Two Cases of Pelvic Fractures Stabilized by External Fixation in Dogs and Classification Proposal"

_vetsci, 2023, doi:10.3390/vetsci10110656_

Round 1
Reviewer 1 Report
Comments and Suggestions for Authors
Thank you very much for submitting this valuable research. This study is a useful paper that addresses postoperative outcomes using external fixation in pelvic fractures, a topic with limited reports in the veterinary field. I have a few concerns, so I kindly request you to make appropriate revisions and reconsider certain aspects. I hope this coment proves helpful.
It appears that the reference number might be incorrect. Could you please correct it?
Lines 51-54. Do you have any references?
Lines 58-59. Are there any reports in the field of veterinary medicine?
Lines 60-62. Should primary and complementary be separated?
Lines 153-156. I think this description should come after the surgical technique.
Lines 195-197. Do you have any references? Also, at what point was the VAS measured?
Lines 203-204. In X-ray images of the pelvis where bones overlap, making evaluation difficult, how do you determine bone healing?
Lines 222-224. Were acetabular fractures other than the four cases and sacroiliac dislocations treated with EF alone?
Lines 238-239. Specifically, where were the lesions located in other parts of their skeletal system, and what kind of impairments and at what frequency were they present?
line 248-250. Did you treat acetabular fractures other than those in the four cases and sacroiliac dislocations with EF alone?
Lines 257-258. Why were these acetabular fractures treated with a combination of internal fixation?
Figure 4 is missing.
Figure 3. Please present the radiographs and explain where the fracture is located.
Figure 5. Please explain where the fracture is located.
Figure 6. Please explain where the fracture is located. Additionally, in this case, was plate fixation applied to the right illiac fracture and lag screw fixation to the left sacroiliac dislocation? Please specify all treatments applied.
Figure 7. Please explain where the fracture is located.
Lines 299-300. Were all these cases performed exclusively under fluoroscopy? Were there any intra-articular fractures?
Lines 307-309. Why did femoral head necrosis occur in this case? Was there a histopathological examination conducted?
Lines 315-316. Was it challenging to install anti-rotational screws or pins as internal fixation during sacroiliac screw placement?
Lines 318-323. How were the decisions made regarding these different treatment methods? I would like an explanation.
Lines 347-348. What is the basis for this statement?
Lines 364-366. Bilateral scores are better than unilateral, but I think it's often the opposite. Due to the small sample size, I don't believe we can make definitive conclusions, but I would like you to consider the reasons behind this.
Lines 367-368. I believe this is because plate fixation is superior to EF. The drawbacks of using EF should be considered. Additionally, I think it's important to mention that this study is limited to the short term, as you have stated in lines 380-383.
Lines 386-410. Please also mention as a limitation that long-term outcomes have not been tracked.
Author Response
Dear Reviewer,
In this revision process, we have attempted to incorporate all the modifications recommended. We hope that the outcome meets your expectations.
We appreciate your attention.
Best regards,

Reviewer 2 Report
Comments and Suggestions for Authors
please find Word file with comments and suggestions attached

needs some english editing, overall OK
Author Response

(The authors gave the same response as above.)

Round 2
Reviewer 1 Report
Comments and Suggestions for Authors
Line 225. Below are the questions I asked last time as well. We would very much appreciate your answers. In X-ray images of the pelvis where bones overlap, making evaluation difficult, how do you determine bone healing?
Line 346. Below are the questions I asked last time as well. We would very much appreciate your answers. Why did femoral head necrosis occur in this case? Was there a histopathological examination conducted?
Line 351-352. Below are the questions I asked last time as well. We would very much appreciate your answers. Was it challenging to install anti-rotational screws or pins as internal fixation during sacroiliac screw placement?
Line 383-384.I do not agree with this opinion, since we end up using metal.
Line 407-410. I don't think that fixation with EF will provide complete repair for an intra-articular acetabular fracture. I think you should mention the progression of osteoarthritis.
Author Response
Dear reviewer,
I appreciate the time invested on the revision of our work. I hope that the changes made are to your liking.
Kind regards,
José A. Flores

Reviewer 2 Report
Comments and Suggestions for Authors
(Last) reviewer’s comments on vetsci-2653729
Thank you for the revision and explanations; in this reviewer’s view, the manuscript can now be recommended for publication, pending some minor corrections listed below:
Please do not use any slang, e.g. “wasn’t” say “was not”; check entire manuscript for possible use of other slang expressions such as “isn’t”.
“…Eight patients (25.00%)…” expressing any value in a sample size of only 32 in tenth or hundreds of a percent is per se nonsense, does not add to precision and may create a false impression of precision (1% is already a hundreds of an entity); the same is true for the outcome reporting such as: “The outcome graded by VAS in the patients with unilateral fractures was excellent in five cases (62.50%), good in two (25.00%), and fair in one (12.50%)”. If the sample size were several hundred, then perhaps outcome may be reported that way but only if the assessment method were in relation to the numerically expressed precision; yet, what you report here are visually assessed estimates of lameness/pain (“patient comfort” – whatever you mean by that ?), expressed as (arbitrary) numeric scores from 1 to 10; expressing results of such subjective assessment method (VAS) in tenth and hundreds of a percent is difficult to accept and, in fact creates (in the inattentive reader) the impression of high “scientific” accuracy and precision – which in fact is methodically lacking.
Sorry for pointing this out in the 2nd review and not in the first place; since this is an oversight on my part, I am not requesting a second major revision, just asking to kindly adjust the numbers by eliminating unnecessary fractions of percentage values and for instance avoiding expressing the result of one single case as 12,50%, for instance (just say in one of 32).
Otherwise, congratulations for the good overall outcomes in these difficult surgical cases; recommend publication after these minor adjustments.
Comments on the Quality of English Language
eliminate slang expressions from text such as "wasn't"
Author Response

(The authors gave the same response as above.)
